# Neurodevelopmental Impact of Maternal Immune Activation and Autoimmune Disorders, Environmental Toxicants and Folate Metabolism on Autism Spectrum Disorder

**DOI:** 10.3390/cimb47090721

**Published:** 2025-09-04

**Authors:** George Ayoub

**Affiliations:** Psychology Department, Santa Barbara City College, Santa Barbara, CA 93109, USA; neuro@sbcc.edu

**Keywords:** autism spectrum disorder, maternal immune activation, environmental exposures, autoimmune disease, cerebral folate deficiency

## Abstract

Autism Spectrum Disorder (ASD) is a complex neurodevelopmental condition characterized by deficits in social communication, repetitive behaviors, and sensory sensitivities. While genetic factors contribute significantly to ASD risk, a growing body of evidence implicates environmental exposures and immune-mediated mechanisms in the etiology and severity of ASD. This review synthesizes peer-reviewed findings on (1) maternal immune activation, (2) environmental toxicant co-exposures, (3) maternal autoimmune disease, and (4) cerebral folate deficiency (via folate receptor alpha autoantibodies), detailing their mechanistic contributions to core and associated ASD symptoms. Collectively, these findings illuminate converging neuroimmune and metabolic pathways that, when disrupted in utero, substantially alter the developmental trajectory of the brain and increase the likelihood of ASD. Such interruptions leading to developmental changes can trigger immune activation from environmental sources of infection and pollution, with these triggers compounded in cases of autoimmune disease or cerebral folate deficiency. Understanding these mechanisms provides a foundation for early identification, stratified risk assessment, and the development of targeted prenatal interventions. Thus, a lesson we learn from autism is that neurodevelopmental disorders should be understood as the product of combined genetic vulnerabilities and modifiable prenatal and postnatal influences. Further exploration of this framework will open paths for precision intervention and prevention.

## 1. Introduction

Autism spectrum disorder (ASD) is a multifactorial neurodevelopmental condition arising from the interplay of genetic, immunological, and environmental factors. In recent years, ASD prevalence has risen globally [1], as have advances in neuroimmunology and translational neuroscience, which have significantly deepened our understanding of the non-genetic contributors to ASD risk and pathogenesis. Evidence now implicates not only maternal immune activation (MIA) from infection or inflammation during pregnancy, but also maternal autoimmune disease, environmental toxicant exposure, and cerebral folate deficiency mediated by folate receptor alpha autoantibodies in the etiology and symptomatology of ASD [2,3,4]. Each of these factors exerts its influence through unique, yet overlapping, biological mechanisms that disrupt fetal neurodevelopment, including aberrant cytokine signaling, chronic neuroinflammation, oxidative stress, mitochondrial dysfunction, synaptic dysfunction, and direct interference with key neurodevelopmental pathways [5,6,7,8,9].

MIA describes the pro-inflammatory maternal responses, often to infections, experienced during pregnancy, which increase levels of cytokines. These cytokines cross the placenta, trigger chronic inflammation in the fetal brain, and alter essential processes of neurodevelopment, contributing to the emergence of neurodivergent behaviors. Beyond acute immune challenges, mothers with chronic autoimmune diseases demonstrate a heightened baseline of systemic inflammation and may produce autoantibodies capable of crossing the placental barrier to directly interact with fetal brain proteins. Such maternal autoantibody transfer is increasingly recognized as a key mechanism for synaptic and neurodevelopmental dysregulation [5,6,7,8,9].

Simultaneously, environmental exposures, including heavy metals, air pollution, and pesticides, can amplify immune activation and increase oxidative stress. These pollutants interact with genetic susceptibilities, further disturbing neuroimmune homeostasis and increasing ASD risk [8,10]. Additionally, autoantibodies directed against folate receptor alpha (FRα) can impede folate transport into the brain, causing cerebral folate deficiency even when peripheral levels appear adequate [11]. Neurological sequelae of this deficiency, including impaired myelination, disrupted neurotransmitter synthesis, and a cluster of behavioral symptoms, are frequently observed in ASD and can respond to targeted metabolic interventions [12].

This report synthesizes the evidence on these four converging themes—maternal immune activation, maternal autoimmune disease and autoantibody transfer, environmental co-exposures, and FRα autoantibody-mediated cerebral folate deficiency, to elucidate their individual and overlapping contributions to ASD pathogenesis. By addressing shared inflammatory, immune, and metabolic mechanisms, this integrated discussion aims to inform early identification, risk stratification, and the development of precision therapies for ASD.

## 2. Maternal Immune Activation (MIA) and ASD Symptomatology

Maternal immune activation (MIA), triggered by infection or inflammation during pregnancy, is increasingly recognized as a key risk factor for neurodevelopmental disorders such as autism spectrum disorder (ASD) [2,3,4]. MIA leads to the release of pro-inflammatory cytokines (notably IL-6, IL-17A, TNF-α), which can traverse the placenta, disturb fetal brain development, and ultimately disrupt critical neurodevelopmental processes including neuronal migration, synaptic formation, and synaptic pruning. The cytokines activate fetal microglia, leading to chronic neuroinflammation, oxidative stress, and mitochondrial dysfunction. These disruptions can manifest as core ASD symptoms, including social withdrawal, cognitive rigidity, and heightened anxiety [5,6,7,8,9].

Maternal immune activation (MIA), which presents as high immune load during pregnancy, is a well-established environmental risk factor for neurodevelopmental disorders such as autism spectrum disorder (ASD). MIA disrupts fetal brain development, leading to autism symptoms like social withdrawal, cognitive rigidity, and heightened anxiety through neuroinflammation, cytokine signaling, and altered neural connectivity. This review examines the mechanisms by which MIA leads to these core autism symptoms.

### 2.1. Immune Activation and Cytokine Signaling

MIA is typically caused by maternal infection or inflammation during pregnancy. Maternal infection or inflammation results in the upregulation of pro-inflammatory cytokines, which either directly enter fetal circulation or activate fetal immune cells. Cytokines such as IL-6 and IFN-γ have been found at elevated levels in ASD patients and animal models following MIA [5,6,13]. The cross-talk between these cytokines appears to modulate MIA’s impact on the developing brain, promoting chronic neuroinflammation and altering neurodevelopmental trajectories [9,14]. Microglia, the brain’s resident immune cells, are activated via cytokine signaling, leading to chronic inflammation, oxidative stress, and mitochondrial dysfunction in the fetal brain. These factors disrupt normal neurodevelopmental processes such as neuronal migration, synapse formation, and pruning [5,6,15].

### 2.2. Neurodevelopmental Changes

MIA leads to abnormal synaptic pruning and connectivity, especially in regions critical for social behavior (prefrontal cortex, amygdala, hippocampus) [16,17]. This altered synaptic connectivity creates a disruption in the balance between excitatory (glutamatergic) and inhibitory (GABAergic) neurotransmission, often due to the reduced function of parvalbumin-positive neurons. This imbalance underlies symptoms such as cognitive rigidity and repetitive behaviors, hallmarks of ASD [15,16,18,19,20,21,22].

In confirmation that it is the viral infection that leads to MIA and not an antiviral treatment, a recent study found that neuropsychiatric events requiring hospitalization were increased with childhood influenza, and an antiviral treatment suppressed this increase [23].

MIA reduces levels of brain-derived neurotrophic factor (BDNF), impairing synaptic plasticity and learning, which contributes to the cognitive rigidity [5,24].

### 2.3. Molecular Factors

Recent work shows the role of the P2X7 receptor signaling pathway in mediating MIA’s effects on ASD through mechanisms involving mitochondrial dysfunction and oxidative stress, further expanding potential molecular targets for intervention [25]. Transcriptomic analyses in animal models have revealed that MIA leads to the downregulation of ASD-risk genes, including those with high penetrance such as FMR1 and CHD8, and strongly affects pathways involved in translation initiation (e.g., the mTOR–EIF4E axis), which are also dysregulated in ASD brains [17]. Human 3D brain organoid models are now being used to study the direct effects of MIA, confirming that cytokines such as IL-6 and IL-17A are necessary and sufficient to induce neurodevelopmental and behavioral abnormalities reminiscent of ASD [16].

Multiple cohort studies confirm an association between maternal infection during pregnancy and increased risk of ASD in offspring, emphasizing the impact of systemic maternal inflammation and elevated midgestational cytokine levels as relevant biomarkers for ASD risk [8,14,26].

### 2.4. Symptom-Specific Pathways Affected by MIA

Three ASD symptoms can be linked to MIA: social withdrawal, cognitive rigidity, and heightened anxiety. The presumed mechanisms for the development of each of these symptoms is detailed below, and summarized in Table 1.

#### 2.4.1. Social Withdrawal

Disrupted social circuits. MIA impairs the development of neural circuits involved in social behavior, such as the prefrontal cortex and amygdala. Animal models show reduced social interaction and communication, mirroring human ASD symptoms [5,18].Cytokine Mediation. IL-6 and IL-17A are particularly implicated; blocking these cytokines in animal models prevents social deficits [5].

#### 2.4.2. Cognitive Rigidity

Impaired Synaptic Plasticity: MIA-induced neuroinflammation disrupts synaptic plasticity, leading to inflexible behavior and difficulty adapting to change (cognitive rigidity).Altered Prefrontal Cortex Function: Changes in inhibitory interneuron function (especially parvalbumin cells) in the prefrontal cortex reduce cognitive flexibility [15,29].

#### 2.4.3. Heightened Anxiety

Amygdala Hyperactivity: MIA increases reactivity in the amygdala, a key region for anxiety and stress responses.Neurotransmitter Imbalance: Delayed GABAergic maturation and altered serotonin/noradrenaline signaling contribute to anxiety-like behaviors [27,28].Behavioral Evidence: Offspring of MIA-exposed mothers show increased anxiety in behavioral tests (e.g., elevated plus maze, open field), and heightened physiological stress responses [28].

## 3. Environmental Co-Exposures: Pollutants and Infections

Environmental co-exposures like pollutants and infections can worsen autism symptoms, trigger earlier onset, and increase comorbid allergies/asthma by disrupting immune, epigenetic, and neurodevelopmental processes.

### 3.1. Immune System Dysregulation

Environmental pollutants (microplastics, heavy metals, air or water pollution, pesticides) and infections can activate the maternal and fetal immune systems, leading to chronic inflammation. This immune activation increases pro-inflammatory cytokines (namely IL-6, IL-17A, TNF-α), which can cross the placenta and disrupt fetal brain development [8,10]. Elevated IL-17A and IL-6 have been implicated in initiating neuroinflammation, activating microglia, and disrupting synaptic development in the fetal brain [5,10,30]. Recent animal and human studies confirm that maternal cytokine surges, particularly IL-17A, lead to structural cortical abnormalities and ASD-like social and behavioral deficits in offspring [10]. MIA can also induce a deficiency in regulatory T cells in the offspring, perpetuating brain inflammation and ASD-like phenotypes—a process reversible via immunomodulatory interventions targeting regulatory T cells [30].

Immune dysregulation is linked to both increased autism symptom severity and the development of comorbid allergic conditions (asthma, eczema), as the same cytokines and immune pathways are involved in neurodevelopment and allergic responses [31,32,33].

### 3.2. Epigenetic and Genetic Interactions

Environmental exposures can cause epigenetic changes (such as DNA methylation, histone modification) in genes critical for brain development and immune function. These changes can increase the risk of earlier symptom onset and more severe autism by altering gene expression during critical periods of neurodevelopment. Children with certain genetic susceptibilities may be more vulnerable to environmental insults, leading to gene–environment interactions that amplify risk [31,34,35].

### 3.3. Environmental Chemical Exposures, Oxidative Stress, and Mitochondrial Dysfunction

Environmental pollutants (heavy metals, air pollution, pesticides, microplastics) and systemic inflammation increase oxidative stress in developing brains [31,36,37]. Elevated reactive oxygen species (ROS) cause oxidative damage to neuronal structures, impairing mitochondrial bioenergetics and contributing to chronic neuroinflammation in ASD [5,37,38]. Studies in ASD patients and animal models reveal reduced antioxidant capacity, notably decreased glutathione (GSH) levels and a lower GSH/GSSG ratio, correlating with increased neuronal damage and synaptic dysfunction [37,38]. Mitochondrial dysfunction, including disrupted electron transport chain activity and energy metabolism, is now recognized as a core pathophysiological feature in ASD, with genetic and environmental factors jointly influencing vulnerability [38,39,40].

Recent epidemiological and meta-analytic evidence associates prenatal exposure to pollutants (nitrogen dioxide, copper, phthalates, pesticides, heavy metals) with increased ASD risk and symptom severity [32]. Environmental co-exposures amplify immune dysregulation and exacerbate oxidative stress, particularly in genetically susceptible children, reinforcing the need to consider both individual and cumulative exposures [32,36]. Epigenetic modifications in response to pollutants (e.g., altered DNA methylation of neurodevelopmental and immune genes) may mediate early onset and increased severity of ASD symptoms [36]. Microbiome alterations secondary to environmental factors also modulate immune development and ASD risk [41].

Microplastics are an increasing source of environmental pollutants that enter the body from food, drink and air, exposing developing bodies to the myriad chemicals [42,43,44]. Ongoing animal studies in a murine model are confirming that exposures to microplastics in general, and to specific components of these plastics in particular, are directly correlated with certain ASD symptoms. Bisphenol A interrupts gene expression in neurons that are associated with ASD behaviors [45]. Mice exposed prenatally to polyethylene showed it accumulated in the brain and created ASD traits in the offspring [46]. And polystyrene nanoplastic exposure caused ASD symptoms of anxiety and depressive behavior in mice, in a process that impaired synaptic transmission in the prefrontal cortex by decreasing the expression of an astrocyte glutamate transporter, and these effects can be reduced by activating the glutamate transporter [47]. These studies identify just three of the thousands of chemicals found in microplastics, raising concern that the abundance of these compounds may have large effects on fetal development.

Systematic reviews confirm maternal infections and pollutant exposures during pregnancy as significant risk factors for ASD across diverse populations [8,32]. Interactions between maternal health (fever, nutrition) and immune activation affect fetal resilience and modify ASD risk and outcomes [10,36].

Thus, oxidative stress damages neurons, impairs synaptic development, and disrupts neurotransmitter systems, contributing to more severe and earlier-appearing autism symptoms. Mitochondrial dysfunction, often observed in children with autism, can be exacerbated by environmental exposures, further impairing brain energy metabolism [31,32].

### 3.4. Microbiome Disruption

Both pollutants and infections can alter the maternal and infant gut microbiome. Microbiome changes affect immune system development and can increase susceptibility to allergies and asthma, as well as influence neurodevelopmental outcomes [31].

### 3.5. Neurodevelopmental Changes Related to Environmental Exposures

Altered Synaptic Connectivity: Environmental exposures disrupt the formation and pruning of synapses, especially in brain regions involved in social behavior and communication.Impaired Myelination: Some pollutants interfere with the development of myelin, slowing neural transmission and affecting cognitive and behavioral functions.Neuroinflammation: Chronic activation of microglia (brain immune cells) leads to persistent neuroinflammation, which is associated with more severe autism symptoms and earlier onset [31,32] (Table 2).

These findings highlight the importance of reducing environmental exposures during pregnancy and early childhood to lower the risk and severity of autism and related comorbidities.

## 4. Maternal Autoimmune Disease (e.g., SLE, Hashimoto’s) and ASD

Maternal autoimmune diseases, including systemic lupus erythematosus (SLE), Hashimoto’s thyroiditis, and rheumatoid arthritis can cause autism symptoms by exposing the fetus to autoantibodies and inflammatory cytokines that cross the placenta, targeting fetal brain proteins and triggering neuroinflammation. New cohort studies and mechanistic research from 2023 to 2025 underscore how maternal immune dysregulation, chronic inflammation, and autoantibody transfer directly impact fetal brain development and behavioral outcomes. This disrupts neural development, synaptic formation, and neurotransmitter balance [4,49].

### 4.1. Population-Based and Meta-Analytic Evidence

Large-scale analyses confirm that maternal autoimmune disease confers a significantly higher risk of ASD in offspring, regardless of underlying diagnosis [26,50]. Conditions such as Sjögren’s syndrome and rheumatoid arthritis are particularly associated with increased risk. Recent meta-analyses estimate about 30% higher odds of ASD among children born to mothers with autoimmune conditions [26,50]. Population-based studies across Taiwan, Europe, and North America consistently replicate these findings, adjusting for confounders such as maternal age and other health conditions [50,51].

### 4.2. Autoantibody Transfer and Fetal Brain Impact

Mothers with autoimmune diseases frequently harbor brain-reactive antibodies, which can cross the placenta during pregnancy and bind to specific fetal brain proteins (such as lactate dehydrogenase A/B, CRMP1/2, STIP1, RPL23, GAPDH, CAMSAP3) [50,52,53]. Experimental models show that maternal IgG autoantibodies—injected during gestation—induce ASD-like social, communication, and behavioral impairments in mice and primates [50,54,55]. Novel autoantibodies against fetal neuronal antigens have recently been identified in a substantial subset of mothers with autistic children, and their presence is considered a candidate biomarker for “Maternal Autoantibody-Related (MAR) Autism” [52,56]. Offspring exposed to these autoantibodies display reduced dendritic spines, altered cortical thickness, and disrupted connectivity in key brain regions involved in social and cognitive functions [4,54,55].

### 4.3. Chronic Maternal Inflammation and Cytokine Exposure

Autoimmune disease in pregnancy is characterized by elevated pro-inflammatory cytokines, particularly IL-6, IL-17A, TNF-α, and IFN-γ [2,9,50,56,57]. These cytokines cross the placenta and initiate neuroinflammatory pathways, activating microglia and disrupting synaptic pruning. Recent research shows that maternal autoantibody patterns are linked with distinct pro-inflammatory cytokine and chemokine profiles, especially increases in interferon-gamma. This immune state is associated with greater ASD severity in the offspring [5,9,56,58]. Regulatory T cell deficiency imprinted during maternal immune activation perpetuates brain inflammation and exacerbates ASD-like phenotypes in animal models [30].

### 4.4. Epigenetic and Neurodevelopmental Changes

Exposure to maternal autoantibodies and inflammation can trigger epigenetic changes in fetal neural cells, affecting gene expression programs crucial for neurodevelopment and social communication [53,58]. These processes may interact with genetic susceptibility, leading to persistent alterations in neural circuits and ASD symptomatology [53,58]. These changes can lead to abnormal neural circuit formation, especially in regions involved in social behavior (prefrontal cortex, amygdala), repetitive behaviors (basal ganglia), and sensory processing (thalamus, sensory cortices) [4,5].

### 4.5. Clinical Implications

Identification of autism-specific maternal antibodies enables risk stratification and potential early biomarkers for ASD [30,52]. Intervention strategies under investigation include immunomodulation and targeting maternal inflammation during critical windows of pregnancy [2].

Maternal autoimmune diseases like SLE and Hashimoto’s can expose the fetus to autoantibodies and inflammatory cytokines, leading to neuroinflammation, disrupted synaptic development, and abnormal neural circuit formation. These changes underlie core autism symptoms such as social communication deficits, repetitive behaviors, and sensory sensitivities, as supported by animal and human studies (Table 3).

## 5. Cerebral Folate Deficiency (CFD) via Folate Receptor Alpha Autoantibodies

Cerebral folate deficiency caused by folate receptor alpha (FRα) autoantibodies disrupts brain folate transport, leading to language delay, poor attention, and sometimes seizures by impairing neurodevelopment and neurotransmitter synthesis.

### 5.1. Folate Receptor Alpha Autoantibodies and Brain Folate Transport

FRα autoantibodies block the folate receptor on the choroid plexus, impairing the transport of 5-methyltetrahydrofolate (5-MTHF) into the brain despite normal blood folate levels. This leads to cerebral folate deficiency (CFD), despite plasma folate levels being normal [49,59,60,61].

Folate is critical for DNA synthesis, myelin formation, and neurotransmitter production. CFD thus disrupts development by impairing myelination, disrupting neurotransmitter synthesis (dopamine, serotonin, GABA), and altering neuronal stability [62,63]. A high prevalence of FRα autoantibodies is observed in ASD children, with affected individuals displaying increased rates of language delay, poor attention, seizures, and adaptive functioning deficits [61,64]. This indicates that FRα autoantibodies are a biomarker for ASD.

### 5.2. Prevalence and Clinical Findings

Approximately 33–70% of children with ASD test positive for FRα autoantibodies, a rate substantially higher than in typically developing children [60,61,65]. Recent clinical cohorts show that FRAA-positive ASD children have significantly poorer communication and adaptive behavior scores [61]. Detection of FRα autoantibodies offers a precision diagnostic tool and helps identify a modifiable metabolic subtype of ASD [60,61].

Animal studies confirm that maternal or early-life exposure to FRα autoantibodies results in ASD-like behavioral and cognitive deficits, including anxiety, social impairment, and learning difficulties. These findings reinforce the pathogenicity of antibody-mediated folate deficiency during neurodevelopment [66]. Human studies document white matter abnormalities, delayed myelination, and immune activation in CFD cases, correlating with ASD-related symptoms [60,67].

### 5.3. Neurodevelopmental Consequences

#### 5.3.1. Language Delay

Folate is essential for myelination, DNA synthesis, and neurotransmitter production (e.g., dopamine, serotonin). Deficiency disrupts myelin development, impairing the development of language-related brain regions (e.g., Broca’s and Wernicke’s areas), which leads to language delay [11,68].

#### 5.3.2. Poor Attention

Folate is critical for monoamine neurotransmitter synthesis (dopamine, norepinephrine, serotonin), which regulate attention and executive function. CFD disrupts these pathways, resulting in poor attention and focus [11].

#### 5.3.3. Seizures

Folate is necessary for GABA and glutamate balance and for maintaining neuronal membrane stability. CFD increases neuronal excitability, predisposing to seizures in some children [11,59].

### 5.4. Treatment and Clinical Correlates

Folinic acid supplementation (rather than commonly used folic acid, which is the oxidized version of folate) can bypass FRα blockade, leading to significant improvements in communication, attention, and social functioning in ASD children positive for FRAA [60,69,70,71]. Meta-analyses now support the routine screening for FRAA in children diagnosed with ASD and consideration of folinic acid therapy in positive cases [60,63].

Additionally, a recent case study reports that treatment during pregnancy with folinic acid may avert the development of autism in the gestated child [72]. In this report, the women receiving folinic acid had previous births that were ASD, and the folinic acid they received during pregnancy may have been the factor that reduced risk in the new birth.

The titer of FRα autoantibodies inversely correlates with CSF 5-MTHF levels, supporting FRα autoimmunity as the cause of CFD in a large subset of ASD cases [60,73]. Delineation of binding vs. blocking FRAA allows the identification of ASD subgroups for targeted intervention [74,75].

### 5.5. Additional Neurodevelopmental Changes

White matter abnormalities and delayed myelination are seen in CFD, affecting cognitive and motor development. Neuroinflammation may be triggered by the immune response to FRα, further impairing brain function.

Cerebral folate deficiency due to folate receptor alpha autoantibodies blocks folate entry into the brain, disrupting myelination, neurotransmitter synthesis, and neuronal stability. This leads to language delay, poor attention, and, in some cases, seizures—symptoms frequently seen in autism spectrum disorder (Table 4).

## 6. Comprehensive Summary

This review highlights a central principle: autism spectrum disorder emerges from the interplay of genetics with immune, environmental, and metabolic stressors during critical periods of brain development. The specific factors, including maternal infections, autoimmune activity, pollutant exposure, and folate transport disruption, are diverse, but they converge on shared neuroimmune pathways that destabilize fetal brain development, as shown in Table 5.

The four factors, MIA, environmental exposures, maternal autoimmune disease, and FRα-mediated cerebral folate deficiency, share overlapping mechanisms: chronic neuroinflammation, oxidative stress, mitochondrial dysfunction, abnormal synaptic development, impaired myelination, and altered neurotransmission. These disruptions alter brain developmental trajectories and increase ASD risk. These factors are summarized in Table 5, and their interactions on development are visualized in Figure 1.

## 7. Conclusions

The convergence of evidence from epidemiological, clinical, and mechanistic studies underscores the central role of maternal immune activation, autoimmune disease with fetal autoantibody exposure, environmental neurotoxicants, and cerebral folate deficiency in shaping autism spectrum disorder (ASD) risk and phenotype. As detailed in this report, maternal infection and inflammation during pregnancy trigger the release of pro-inflammatory cytokines that can cross the placenta, disrupt fetal neurodevelopment, and lay the groundwork for chronic neuroinflammation, oxidative stress, and synaptic dysfunction, leading to ASD symptomatology. Similarly, maternal autoimmune disease and the presence of brain-reactive autoantibodies further contribute to altered neural connectivity and increased ASD risk through direct effects on the developing fetal brain.

Environmental exposures to microplastics, heavy metals, air pollution, and pesticides amplify immune dysregulation and oxidative stress, especially in genetically predisposed fetuses. The discovery of folate receptor alpha autoantibodies has provided novel insight into how impaired folate transport across the blood–brain barrier creates cerebral folate deficiency, leading to impaired myelination and behavioral deficits relevant to ASD, with recent studies confirming the efficacy of targeted metabolic intervention, and providing a non-toxic therapeutic with significant benefit to a large fraction of ASD children.

Collectively, these findings reveal overlapping neuroimmune and metabolic pathways that, when dysregulated by prenatal exposures, can significantly alter key stages of brain development and result in long-lasting neurobehavioral consequences. They also highlight the importance of early identification, risk stratification, and targeted interventions for at-risk populations. Continued research on these interacting risk factors and biologically informed treatment approaches holds promise for advancing personalized care and improving outcomes for individuals with ASD.

## Figures and Tables

**Figure 1 cimb-47-00721-f001:**
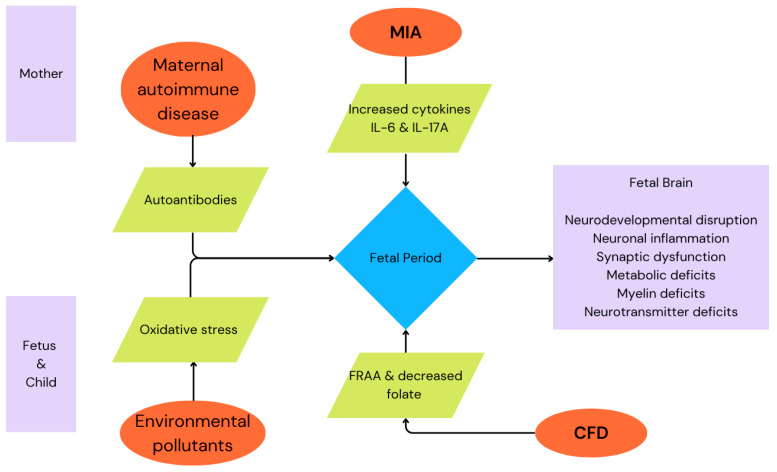
Summary schematic of the four factors described, and their impact on fetal development. Maternal factors are above, fetal and child factors are below. As explained in the text, MIA and environmental pollutants compound oxidative stress on the fetus during critical periods of development. These stresses can be further compounded in cases of cerebral folate deficiency. The result is inflammation in the fetal brain which alters development, creating the dysfunction of synaptogenesis and pruning, deficits in myelination and neurotransmission, leading to ASD-like conditions in the child.

**Table 1 cimb-47-00721-t001:** Summary of key MIA mechanisms in triggering ASD symptoms.

Symptom	Mechanism Involved	Impact on Fetal Brain/ASD
Increased ASD risk	Maternal infection/systemic inflammation	Increased ASD risk in population studies [8,14,26]
Social Withdrawal	Pro-inflammatory Cytokines (IL-6, IL-17A, IFN-γ), impaired connectivity	Microglial activation, neuroinflammation, synaptic dysfunction [5,9,14,16]
Cognitive Rigidity	Abnormal synaptic pruning, excitatory/inhibitory imbalance	Cognitive, social and behavioral deficits [16,17,19,24]
Heightened Anxiety	Oxidative stress, mitochondrial dysfunction	Impaired energy metabolism, neuronal stress [5,25,27,28]
Cognitive Dysfunction	Gene expression dysregulation (e.g., FMR1, CHD8, mTOR)	Altered neurogenesis, synaptic connectivity [16,17]

**Table 2 cimb-47-00721-t002:** Symptom-specific outcomes related to environmental triggers.

Mechanism	Role in ASD Pathogenesis	Reference
Pro-inflammatory cytokines (IL-6, IL-17A)	Disrupted synaptic development, neuroinflammation, microglial activation	[5,8,10,30]
Oxidative stress, mitochondrial dysfunction	Neuronal injury, reduced synaptic plasticity, energy impairment	[5,37,38,39,48]
Environmental pollutants	Immune activation, oxidative/epigenetic changes, altered brain development	[32,36]
Gene–environment and epigenetic interactions	Exacerbated ASD risk and severity in genetically susceptible individuals	[32,36]
Microbiome disruption	Modified immune and neural maturation	[41]
Maternal infection/systemic inflammation	Increased ASD rates, biomarker identification, dose–response effects	[8,10,32]

**Table 3 cimb-47-00721-t003:** Symptom-specific pathways activated by maternal autoimmune disease.

Mechanism	Impact on Fetus/ASD Pathogenesis	References
Maternal brain-reactive autoantibodies	Direct neuronal binding, synaptic disruption, ASD-like behaviors in animal models	[50,52,54,55]
Chronic maternal inflammation (IL-6, IL-17A, etc.)	Microglial activation, abnormal neurodevelopment, heightened ASD risk	[9,50,56,57]
Epigenetic and gene–environment interactions	Aberrant gene regulation, altered neural circuit assembly	[2,53]
Population risk association	Elevated ASD incidence in offspring of mothers with autoimmune/inflammatory conditions	[26,50,51]

**Table 4 cimb-47-00721-t004:** Symptom-specific pathways impacted by cerebral folate deficiency.

Mechanism	ASD-Associated Outcomes	References
FRα autoantibody-mediated CFD	Language delay, attention deficits, seizures	[60,61]
Impaired brain folate transport	White matter/myelination deficits	[60,67]
FRAA positivity in ASD	High subgroup prevalence, worse outcomes	[60,61]
Folinic acid (not folic acid) supplementation	Improved language, behavior, overall function	[12,60,69,70,71]
Animal modeling of antibody exposure	ASD-like and cognitive phenotypes	[66]

**Table 5 cimb-47-00721-t005:** Four factors in autism development.

Factor	Effect
Maternal Immune Activation (MIA)	Infections or systemic inflammation during pregnancy elevate pro-inflammatory cytokines (IL-6, IL-17A, TNF-α). These cytokines cross the placenta, activate microglia, and disrupt fetal neurodevelopment, resulting in failure of synapses. MIA is strongly linked to core ASD symptoms such as social withdrawal, cognitive rigidity, and heightened anxiety.
Environmental Co-Exposures	Pollutants induce epigenetic changes that amplify ASD severity and interact with genetic vulnerabilities. Pollutants and infections disrupt synaptic connectivity, impair myelination, and alter neurotransmitter systems. These exposures can worsen ASD symptoms and promote comorbidities like allergies and asthma.
Maternal Autoimmune Diseases (SLE, Hashimoto’s, RA, etc.)	Mothers with autoimmune conditions have heightened systemic inflammation. Their autoantibodies cross the placenta and bind to fetal brain proteins, altering neural circuits and synaptic function. Elevated cytokines and altered epigenetic programming further increase ASD severity.
Cerebral Folate Deficiency (CFD) via Folate Receptor Alpha Autoantibodies (FRαA)	FRα autoantibodies block folate transport into the brain, despite normal blood folate. Folate deficiency disrupts myelination, neurotransmitter synthesis, and DNA methylation. Folinic acid (but not folic acid) supplementation bypasses the blockade and improves communication, behavior, and cognitive function.

## Data Availability

No new data were created or analyzed in this study. Data sharing is not applicable to this article.

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
