# Peer review of "Neurodevelopmental Impact of Maternal Immune Activation and Autoimmune Disorders, Environmental Toxicants and Folate Metabolism on Autism Spectrum Disorder"

_cimb, 2025, doi:10.3390/cimb47090721_

Round 1

Reviewer 1 Report

Comments and Suggestions for Authors

This paper included intereating novel findings, howeverm tthe abastract and summarty were poor wieitten.

The main important finding  in the main areas of brain and the asociation between maternal immun activtion and synaptic unction  are more presicely deeded. Futher, the originnal gene of this pathophysiology may be needed.

Author Response

Thank you for your kind comments, and helpful recommendations. I have revised the abstract to help clarify it. I have provided an overview to summarize the overall points now in the summary figure (fig 1), with a figure legend that provides this overview. I also have included a graphical abstract for your review.

To address the association specifics, I have revised section 1  and section 2 to include further citations to the relevant work on MIA and its impact. As for the original gene, there are multiple genes that are involved in the pathophysiology, and I have addressed this in part in a review of molecular mechanisms in ASD (https://www.mdpi.com/1422-0067/26/13/6483). To add it here would add complication to the focus on MIA.

Reviewer 2 Report

Comments and Suggestions for Authors

This review synthesizes current evidence on four key environmental/immune-mediated pathways contributing to autism spectrum disorder (ASD): maternal immune activation (MIA), environmental co-exposures, maternal autoimmune disorders, and cerebral folate deficiency (CFD). It successfully highlights converging neuroimmune-metabolic mechanisms in ASD pathogenesis and emphasizes implications for early intervention. However, several critical issues require attention to strengthen clarity, methodological rigor, and translational impact.

Major Comments

  1. The abstract omits "environmental toxicants" despite their prominence in the paper. The introduction briefly mentions "environmental co-exposures" but lacks specificity on key pollutants.
  2. Sections 2.1 and 2.2 repeat cytokine mechanisms (IL-6, IL-17A) and neurodevelopmental impacts (synaptic pruning, microglial activation) covered in the Introduction. Table 1 also duplicates text.
  3. The term "microplastics" is introduced without epidemiological evidence linking them directly to ASD (only general immune dysregulation is discussed). Claims about "earlier ASD onset" due to epigenetics lack citations.

Minor Comments

  1. Add "E/I imbalance" (excitatory/inhibitory) to the Abbreviations list.
  2. Page 10, Hall et al. (2023) citation is incomplete.

Author Response

Thank you for your kind words and for your recommendations to improve the manuscript. I have address each of the comments as below:

Major comments

  1. I have named environmental toxins in a new sentence in the abstract to address my previous lack.
  2. I have eliminated the redundance in section 1 and 2 by focusing the introduction (section 1) on the overview and keeping the detail of specific cytokines and specific alterations in section 2.
  3. I have added a new paragraph on microplastics (lines 191-203), with 6 new references to address the need for specifics and sufficient citations to the current literature.

Minor comments:

  1. E/I is now in the abbreviations list
  2. The Hall citation in references (page 11) is complete

Reviewer 3 Report

Comments and Suggestions for Authors

Ayoub presented a review updating the neurodevelopmental impact of maternal immune activation, autoimmune disorders, environmental toxicants, and folate metabolism on Autism Spectrum Disorder. Please find my review comments below.

  1. Lines 24-61 (Introduction section): Please cite more references to support your descriptions. With extensive descriptions supported by only one citation, readers may question the credibility of your assertions.
  2. Please specify the selection criteria employed for the literature review, including but not limited to temporal scope, data sources, and inclusion/exclusion parameters.
  3. There was no graphical abstract for this manuscript.
  4. The review is overly descriptive and lacks a comprehensive summary and generalization.
  5. The Conclusion section is too lengthy. Please shorten it.

Author Response

Thank you for recommendations to improve this manuscript. I have addressed each comment as follows:

  1. The introduction section now has a dozen citations to rectify the deficiency
  2. In my literature review, I am biased toward more recent studies (since 2020) when appropriate, with the exception of longstanding evidence. I use PubMed as the primary search tool, and seek to be inclusive of all reports or recent reviews when there is an abundance of reports.
  3. I have corrected the omission, and include a graphical abstract now
  4. I have added a comprehensive summary with Figure 1 and its legend. This provides a generalized overview of the full manuscript
  5. The conclusion is now broken into 2 short paragraphs and a concluding one following the summary

Round 2

Reviewer 1 Report

Comments and Suggestions for Authors

The  main mecanisms   in ASD may be mainly falure of development of neuronal stnapse , specially defect in astrocyte. The papr may be needdd this falire of develop,ent of synapse fumcton.

Ths paper is less important

Author Response

While I agree that failure of development of synapses due to astrocyte defects is a key issue in autism, I do not think the literature supports that it is the main point, as there are other failures that are also key. These include the metabolic deficits and the myelination failure, as well as the overall disruption in neurodevelopmental timing.

To give more support to your point, I have added a section (Comprehensive Summary) where I specifically name failure of synapse formation (in Table 5) as an important factor. This is on pages 10-11 of the revised manuscript. The summary provides a more comprehensive overview of all mechanisms discussed, and their interactions that build to autism are illustrated in Figure 1.

As for the importance of the paper, I do believe it is important to help the community view autism as not one or another specific failure, which is often what I see, but to recognize that a developmental disorder such as autism has multiple factors, and as a spectrum disorder, different factors are creating the variants of autism that we see each day.

Reviewer 2 Report

Comments and Suggestions for Authors

I have no more comments.

Author Response

Thank you

Reviewer 3 Report

Comments and Suggestions for Authors

Dear authors, thank you for your response. However, the review remains overly descriptive and lacks a comprehensive summary and generalization.

Author Response

I chose the descriptive nature of the paper that you reference, in order for the reader to have a broad understanding of the collection of works that relate to autism. I have seen many in our field refer only to work in one area or another, and not synthesize that there are multiple 'fields' examining autism, and all of them are critical to a clear understanding.

I can understand why you want a summary and generalization to help the reader follow this, and thank you for the recommendation.

I have added a new section (Comprehensive Summary) to more fully address your concern. I realize there are multiple factors in play in this manuscript and appreciate the request to add this comprehensive summary to help tie everything together. This summary, along with the added generalization in the abstract (see below), should help the reader both understand the interactions involved and the directions this may provide for autism research. The comprehensive summary is on pages 10-11, and includes a table (Table 5) that lays out the four factors discussed in the paper and the key impact of each. Figure 1, a visualization of these four factors in creating the conditions for autism is now in this section to provide a visual overview for the reader.

I addressed the generalization need by adding 2 sentences to the Abstract to name the generalization lesson the reader can take from this manuscript. They read: “Thus, a lesson we learn from autism is that neurodevelopmental disorders should be understood as the product of combined genetic vulnerabilities and modifiable prenatal and postnatal influences. Further exploration of this framework will open paths for precision intervention and prevention.”

Round 3

Reviewer 1 Report

Comments and Suggestions for Authors This peper did not introduce the importanr role of synapse function in the ASD, The revider version may be needed

Author Response

Thank you for your request to introduce the important role of synapse dysfunction in ASD.

As you know, I recently added it to the comprehensive summary and the new Table 5 (line373), as well as Figure 1 (line 380).

Since you ask to introduce it, I have added a phrase in the introduction to specifically name synapse dysfunction (line 42) as a key aspect in ASD development. Further in the introduction, two paragraphs lay out aspects of synaptic dysfunction: lines 43-51 and lines 52-60.

I can affirm that such synaptic changes as relates to autism are also included in section 2, specifically 2.4.1, lines 130-ff, 2.4.2, lines 137-ff and 2.4.3, lines 146-ff. The Table 2 (line 234) and Table 3 (line 296) include synaptic dysfunction aspects in their summaries of outcomes (Table 2) and pathways (Table 3).

You are right that synaptic dysfunction is a key alteration in the development of autism, and I hope that I have introduced it sufficiently clearly.

Reviewer 3 Report

Comments and Suggestions for Authors

No comments.

Author Response

Thank you